# Registered report: Transcriptional amplification in tumor cells with elevated c-Myc

**David Blum[1], Haiping Hao[2], Michael McCarthy[3], Reproducibility Project: Cancer Biology\*†**

[1]Bioexpression and Fermentation Facility, University of Georgia, Athens, Georgia; [2]JHMI Deep Sequencing and Microarray Core Facility, Johns Hopkins University, Baltimore, Maryland; [3]University of Oxford, Oxford, United Kingdom

## REPRODUCIBILITY PROJECT CANCER BIOLOGY

**Abstract** The Reproducibility Project: Cancer Biology seeks to address growing concerns about reproducibility in scientific research by conducting replications of 50 papers in the field of cancer biology published between 2010 and 2012. This Registered report describes the proposed replication plan of key experiments from 'Transcriptional amplification in tumor cells with elevated c-Myc' by *Lin et al. (2012)*, published in *Cell* in 2012. The experiments that will be replicated are those reported in Figures 3E and 3F. In these experiments, elevated levels of c-Myc in the P493-6 cell model of Burkitt's lymphoma results in an increase of the total level of RNA using UV/VIS spectrophotometry (Figure 3E; *Lin et al., 2012*) and on the mRNA levels/cell for a large set of genes using digital gene expression technology (Figure 3F; *Lin et al., 2012*). The Reproducibility Project: Cancer Biology is a collaboration between the Center for Open Science and Science Exchange, and the results of the replications will be published in *eLife*.

**\*For correspondence:** tim@cos.io

**Group author details**
†Reproducibility Project: Cancer Biology
See page 15

## Introduction

The proto-oncogene *MYC* is frequently amplified in human cancers and encodes the transcription factor c-Myc, which is associated with a variety of cellular processes such as cell growth and proliferation (*Dang, 2013*). While overexpressed c-Myc is known to contribute to tumorigenesis, an understanding of how this process occurs is complicated by a number of issues, including the large number of binding sites and the diversity between systems. While this was thought to occur by regulation of a specific subset of genes, *Lin et al. (2012)* present findings that c-Myc functions by globally amplifying the expression of actively transcribed genes.

The system used is the human P493-6 B cell model of Burkitt's lymphoma, which contains a tetracycline-repressible *MYC* transgene, allowing for titration of c-Myc protein (*Schuhmacher et al., 1999; Pajic et al., 2000*). The levels of c-Myc can be reduced and subsequently re-induced in a gradual time-dependent manner, as determined by western blot, which is shown in Figure 1B (*Lin et al., 2012*). As soon as 1 hr after re-induction, the protein levels of c-Myc increased above the repressed levels, which by 24 hr were similar to the tetracycline-free condition (*Lin et al., 2012*). This system has been used in other studies with similar results observed (*Schuhmacher et al., 1999; Pajic et al., 2000*). This experiment is important to replicate because it assesses the level of c-Myc re-induction with this system that will be used in the following experiments. This experiment is replicated in Protocol 1.

In Figure 3E the total levels of RNA, in P493-6 cells before and after re-induction of c-Myc, was determined by UV/VIS spectrophotometry (*Lin et al., 2012*). *Lin et al. (2012)* reported an increase in levels of absolute RNA over the timecourse of c-Myc re-induction. This experiment shows that c-Myc increases RNA content per cell, indicating c-Myc functions primarily in transcriptional amplification. Related experiments, using mouse B cells, also observed the same c-Myc-dependent amplification of

cellular RNA content (*Nie et al., 2012*; *Sabò et al., 2014*). However, similar experiments in 3T9 fibroblasts or U2OS cells expressing an inducible c-Myc did not observe an increase in cellular RNA content (*Sabò et al., 2014*; *Walz et al., 2014*). Interestingly, an increase in total cellular RNA content was observed in 3T9 fibroblasts following serum stimulation, which was not observed when c-Myc was deleted in these cells (*Sabò et al., 2014*). This experiment is replicated in Protocol 2.

In Figure 3F, *Lin et al. (2012)* examined the transcriptional profile of a large number of genes from multiple functional categories in cells before and after re-induction of c-Myc using digital gene expression. Re-induction of c-Myc increased the number of active genes (defined as greater than 1 transcript/cell), but did not alter silent genes (defined as less than 0.5 transcript/cell) (*Lin et al., 2012*). This key finding suggests that elevated c-Myc levels lead to an amplification of the existing transcriptional profile. This finding was observed in other experiments using mouse B cells treated with the Myc-Max dimerization inhibitor 10058-F4 and analyzed by ChIP-Seq (*Nie et al., 2012*). Another report using P493-6 cells also found an increase in transcription of c-Myc target genes when c-Myc re-induction was titrated to different concentrations by microarray analysis (*Schuhmacher and Eick, 2013*). This experiment is replicated in Protocols 3 and 4. Recently, two papers were published that used primarily RNA-seq and ChIP-seq to focus on assessing if the transcriptional effects of c-Myc are direct or indirect (*Alderton, 2014*). These studies found that RNA amplification and promoter/enhancer invasion by c-Myc were separable events in 3T9 fibroblasts and U2OS cells, suggesting that c-Myc regulates a distinct subset of genes, which indirectly lead to RNA amplification (*Sabò et al., 2014*; *Walz et al., 2014*).

## Materials and methods

### Protocol 1: Western blot of c-Myc reactivation in tetracyclin-repressible system

This experiment tests the expression of the tetracycline (tet)-repressible *Myc* transgene after repression by tet and re-induction by removal into tet-free medium. This is a replication of the data presented in Figure 1B and assesses the levels of c-Myc protein. One major variable in this system is serum, which has been reported to stimulate the expression of a majority of genes independently from c-Myc (*Schlosser et al., 2005*). P493-6 cells will be cultured with two separate lots of serum that will be maintained throughout all the experiments to assess if there is variability between different batches of serum.

### Sampling

- Experiment to be repeated a total of three times.

  1. This replication attempt will be repeated three times to analyze if this effect is detected and to assess the variance of this system, since it will be repeated three times in the other replication protocols.

- Each experiment has two cohorts:

  1. Cohort 1: P493-6 cells cultured in FBS lot #1.
  2. Cohort 2: P493-6 cells cultured in FBS lot #2.

- Each cohort has four conditions:

  1. Cells cultured in tet-free media.
  2. Cells cultured 0 hr after tet induction.
  3. Cells cultured 1 hr after tet induction.
  4. Cells cultured 24 hr after tet induction.

### Materials and reagents

| Reagent | Type | Manufacturer | Catalog # | Comments |
|---|---|---|---|---|
| P493-6 modified Burkitt's lymphoma cells | Cell line | Original lab | | From original lab |

*Table 1. Continued on next page*

*Table 1. Continued*

| Reagent | Type | Manufacturer | Catalog # | Comments |
|---|---|---|---|---|
| RPMI 1640 medium with sodium bicarbonate, without L-glutamine | Cell culture | Sigma–Aldrich | R0883 | No catalog # listed in original paper |
| Tetracycline system approved FBS | Serum | Clontech | 631105 | Two separate lots will be used |
| Ala-Gln (L-glutamine substitute) | Cell culture | Sigma–Aldrich | G8541 | Original lab used GlutaMAX from Invitrogen |
| T75 flask | Labware | Sigma–Aldrich | Z707503 | Originally not specified |
| Tetracycline | Pharmacological agent | Sigma–Aldrich | T7660 | |
| C-chip disposable hemocytometer | Labware | Digital Bio | DHC-N01 | No catalog # listed in original paper |
| Refrigerated centrifuge | Equipment | Eppendorf | 5810R | Original lab used a Sorvall Legend centrifuge |
| PBS, without $MgCl_2$ and $CaCl_2$ | Buffers | Sigma–Aldrich | D8537 | Originally not specified |
| RIPA lysis buffer | Buffers | Sigma–Aldrich | R0278 | |
| SIGMAFAST Protease Inhibitor Tablets | Inhibitors | Sigma-Aldrich | S8820 | Original lab used Halt Protease Inhibitor cocktail from Thermo-Fisher |
| Phosphatase inhibitor cocktail 2 | Inhibitors | Sigma–Aldrich | P5726 | |
| Phosphatase inhibitor cocktail 3 | Inhibitors | Sigma–Aldrich | P0044 | |
| Bradford Reagent | Reporter assay | Sigma–Aldrich | B6916 | Original lab used Bradford Reagent from Bio-Rad |
| TruPAGE LDS sample buffer (4X) | Buffers | Sigma-Aldrich | PCG3009 | Originally not specified |
| ß-mercaptoethanol | Chemicals | Sigma–Aldrich | M3148 | Originally not specified |
| Nuclease-free water | Chemicals | Sigma–Aldrich | W4502 | Original lab used nuclease-free water from Ambion (AM9938) |
| Mini-PROTEAN Electrophoresis System | Equipment | Bio-Rad | | Originally not specified |
| TruPAGE Precast Gels 4–20% | Western materials | Sigma-Aldrich | PCG2012 | Original lab used 4–12% Bis-Tris Midi gels from Invitrogen |
| TruPAGE TEA-Tricine SDS Running Buffer (20X) | Buffers | Sigma–Aldrich | PCG3001 | Originally not specified |
| ECL DualVue Western Markers (15–150 kDa) | Western materials | Sigma–Aldrich | GERPN810 | Originally not specified |
| Hybond P Western blotting membranes; PVDF | Western materials | Sigma–Aldrich | GE10600029 | No catalog # listed in original paper |
| TruPAGE Transfer Buffer (20X) | Buffers | Sigma-Aldrich | PCG3011 | Originally not specified |
| Methanol | Chemicals | Sigma–Aldrich | 494437 | Originally not specified |
| Semi-dry TransBlot SD System | Equipment | Bio-Rad | 170–3940 | |
| 5% milk powder | Western materials | Sigma–Aldrich | M7409 | Originally not specified |
| 10X TBS buffered saline | Buffers | Sigma–Aldrich | T5912 | Originally not specified |
| Tween-20 | Chemicals | Sigma–Aldrich | P1379 | Originally not specified |

*Table 1. Continued on next page*

*Table 1. Continued*

| Reagent | Type | Manufacturer | Catalog # | Comments |
|---------|------|--------------|-----------|----------|
| Rabbit c-Myc | Antibodies | Epitomics | 1472-1 | Dilute 1:5000; 57 kDa |
| Mouse ß-actin | Antibodies | Sigma–Aldrich | A5441 | Dilute 1:10,000; 42 kDa |
| Goat anti-rabbit-HRP | Antibodies | Sigma–Aldrich | GERPN2124 | Originally not specified. Dilute 1:10,000 |
| Goat anti-mouse-HRP | Antibodies | Sigma–Aldrich | GERPN2124 | Originally not specified. Dilute 1:10,000 |
| ECL Prime Chemiluminescent reagent | Western materials | Sigma–Aldrich | GERPN2232 | Originally not specified |
| G:BOX iChemi XT | Equipment | Syngene | | Original lab used a Gel Dox XR+ System from Bio-Rad |
| GeneSnap | Software | Syngene | Version 6.00.19 | Original Lab used Image Lab software from Bio-Rad |

## Procedure

### Notes

- All cells will be sent for mycoplasma testing and STR profiling.
- Tet-free media: RPMI1640 supplemented with 10% tet system approved FBS and 1% Ala-Gln.
- Cells maintained at 37°C in a humidified atmosphere at 5% $CO_2$.
- P493-6 cells are maintained in two separate FBS lots (lot #1 and lot #2).

1. Prepare stocks in upright T75 flasks of P493-6 modified Burkitt's lymphoma cells cultured in tet-free media.
2. On day 0, seed Flask A with ~$1 \times 10^7$ P493-6 cells in 20 ml tet-free media, and Flask B with ~$2 \times 10^7$ P493-6 cells in 40 ml tet-free media with 0.1 µg/ml tet for a concentration of $0.5 \times 10^6$ cells/ml.

   a. Count cells in each flask using a C-chip disposable hemocytometer.

3. 72 hr later, spin cells down from flasks, and wash cells three times in tet-free media.
4. From Flask A cells, seed 1 new flask with $1.2 \times 10^7$ in fresh tet-free media.

   a. Count cells using a C-chip disposable hemocytometer.

5. From Flask B cells, seed three new flasks with $1.2 \times 10^7$ cells in 20 ml of fresh tet-free media.

   a. Count cells using a C-chip disposable hemocytometer.

6. At time points 0 hr (immediately), 1 hr, and 24 hr, harvest protein lysates from $1 \times 10^7$ cells from a Flask B cell suspension.

   a. Count cells in each flask using a C-chip disposable hemocytometer.
   b. To harvest lysates:

      i. Pellet $1 \times 10^7$ cells at 4°C at 1200 rpm for 5 min in a Sorvall Legend centrifuge.
      ii. Wash pellets once with ice-cold 1X PBS.

        - Cell pellets can be snap frozen in liquid nitrogen and stored at −80°C until needed.

      iii. Resuspend pellets in 100 µl RIPA lysis buffer containing 2× SIGMAFAST Protease inhibitors and 2× Phosphatase inhibitor cocktails 2 and 3.

7. At 24 hr, harvest protein lysate from tet-free cells (Flask A) as described in step 6.
8. Quantify total protein concentration in each lysate using the Bradford assay, according to the manufacturer's instructions.

   a. Use a bovine serum albumin standard curve for the quantification.

9. Run an electrophoresis gel under denaturing conditions, loading the same volume of 50 µg of total protein lysate, with 1× SDS loading buffer, 2.5% ß-mercaptoethanol, and nuclease-free water for each condition, on a 4–20% TruPAGE precast gel. Run at 200 V until the dye front reaches the reference line following manufacturer's instructions.

 a. Samples run per gel:

 i. Protein molecular weight marker.
 ii. 0 hr from tet release (FBS lot #1).
 iii. 1 hr from tet release (FBS lot #1).
 iv. 24 hr from tet release (FBS lot #1).
 v. tet-free control (FBS lot #1).
 vi. 0 hr from tet release (FBS lot #2).
 vii. 1 hr from tet release (FBS lot #2).
 viii. 24 hr from tet release (FBS lot #2).
 ix. tet-free control (FBS lot #2).

10. Transfer gel to a PVDF membrane, using the semi-dry TransBlot SD system according to manufacturer's instructions.
11. Block with 5% milk powder in TBST buffer.

 a. TBST buffer: TBS with 0.2% Tween-20.

12. Probe membrane with the following primary antibodies diluted in 5% milk powder in TBST buffer:

 a. rabbit c-myc; use at 1:5000; 57 kDa.
 b. mouse beta-actin; use at 1:10,000; 42 kDa.

13. Wash membranes in TBST buffer.
14. Detect primary antibodies with the following appropriate secondary antibodies diluted in 5% milk powder in TBST buffer.

 a. goat anti-rabbit-HRP; use at 1:10,000.
 b. goat anti-mouse-HRP; use at 1:10,000.

15. Detect signal with chemiluminescent reagent.
16. Image the chemiluminescent signal using a charge-coupled device (CCD) detection system.
17. Repeat independently two additional times.

## Deliverables

- Data to be collected:

1. STR profile and result of mycoplasma testing of P493-6 cells.
2. Images of the chemiluminescence signal from the c-myc and beta-actin probed membrane (full images with ladder). (Compare to Figure 1B).
3. Raw values and bar graphs of mean signal intensities of the bands normalized for beta-actin levels.

## Confirmatory analysis plan

This experiment assesses the relative levels of c-Myc protein after re-induction, which was showed as a representative image in the original paper. The original paper showed an increase in c-Myc as soon as 1 hr after re-induction, which by 24 hr was similar to the tet-free condition. This replication attempt will perform the following statistical analysis listed below for each cohort.

- Statistical Analysis:

Note: At the time of analysis we will perform the Shapiro–Wilk test and generate a quantile–quantile (q–q) plot to assess the normality of the data and also perform Levene's test to assess homoscedasticity. If the data appear skewed, we will perform an appropriate transformation in order to proceed with the proposed statistical analysis. If this is not possible, we will perform the equivalent non-parametric test.

1. One-way ANOVA comparing the normalized c-Myc levels in cells cultured 0 hr, 1 hr, and 24 hr from tet release and tet-free control.

    • Planned comparisons with the Bonferroni correction:

        1. Normalized c-Myc levels in cells cultured 0 hr from tet release compared to cells cultured 24 hr from tet release.
        2. Normalized c-Myc levels in cells cultured in tet-free control medium compared to cells cultured 24 hr from tet release.

• Meta-analysis of effect sizes:

    1. Compare the effect sizes of the two cohorts (separate FBS lots) and use a meta-analytic approach to combine the two cohort effects, which will be presented as a forest plot.

## Known differences from the original study

The replication will only include the time points used in further studies instead of the full time-course shown in Figure 1B. A separate lot of serum was included to assess if there is variability between different batches of serum. All known differences are listed in the materials and reagents section above, with the originally used item listed in the comments section. All differences have the same capabilities as the original and are not expected to alter the experimental design.

## Provisions for quality control

The cell line used in this experiment will undergo STR profiling to confirm its identity and will be sent for mycoplasma testing to ensure that there is no contamination. All of the raw data, including the image files and quantified bands from the western blot, will be uploaded to the project page on the OSF (https://osf.io/mokeb/) and made publically available. This experiment is also the quality control for the other replication protocols as it assesses the level of c-Myc re-induction with this system.

## Protocol 2: total RNA expression during c-Myc re-activation

This experiment tests the effect of elevated levels of c-Myc on the total levels of RNA in P493-6 cells. This is a replication of the data presented in Figure 3E, which assess RNA levels by UV/VIS spectrophotometry.

## Sampling

• Experiment to be repeated a total of three times for a minimum power of 96%.

    1. See Power calculations section for details.

• Each experiment has two cohorts:

    1. Cohort 1: P493-6 cells cultured in FBS lot #1.
    2. Cohort 2: P493-6 cells cultured in FBS lot #2.

• Each cohort has three conditions:

    1. Cells cultured 0 hr after tet induction.
    2. Cells cultured 1 hr after tet induction.
    3. Cells cultured 24 hr after tet induction.

## Materials and reagents

| Reagent | Type | Manufacturer | Catalog # | Comments |
|---|---|---|---|---|
| P493-6 modified Burkitt's lymphoma cells | Cell line | Original lab | | From original lab |
| RPMI 1640 medium with sodium bicarbonate, without L-glutamine | Cell culture | Sigma–Aldrich | R0883 | No catalog # listed in original paper |

*Table 2. Continued on next page*

*Table 2. Continued*

| Reagent | Type | Manufacturer | Catalog # | Comments |
|---------|------|-------------|-----------|----------|
| Tetracycline system approved FBS | Serum | Clontech | 631105 | Two separate lots will be used |
| Ala-Gln (L-glutamine substitute) | Cell culture | Sigma–Aldrich | G8541 | Original lab used GlutaMAX from Invitrogen |
| T75 flask | Labware | Sigma–Aldrich | Z707503 | Originally not specified |
| Tetracycline | Pharmacological agent | Sigma–Aldrich | T7660 | |
| C-chip disposable hemocytometer | Labware | Digital Bio | DHC-N01 | No catalog # listed in original paper |
| Refrigerated centrifuge | Equipment | Eppendorf | 5810R | Original lab used a Sorvall Legend centrifuge |
| TRI reagent | Chemical | Sigma–Aldrich | T9424 | Included during communication with original authors. Original lab used Trizol |
| 1-Bromo-3-chloropropane | Chemical | Sigma–Aldrich | B9673 | Included during communication with original authors |
| miRVana miRNA extraction kit | Buffer | Ambion | AM1561 | Original lab used #AM1560 with phenol, which is not used in protocol |
| Nuclease-free water | Chemicals | Sigma–Aldrich | W4502 | Original lab used nuclease-free water from Ambion (AM9938) |
| NanoDrop UV/Vis Spectrophotometer | Equipment | Thermo Scientific | ND-1000 | |
| NanoDrop Operating | Software | Thermo Scientific | Version 3.3 | |

## Procedure

- All cells will be sent for mycoplasma testing and STR profiling.
- Tet-free media: RPMI1640 supplemented with 10% tet system approved FBS and 1% Ala-Gln.
- Cells maintained at 37°C in a humidified atmosphere at 5% $CO_2$.
- P493-6 cells are maintained in two separate FBS lots (lot #1 and lot #2).

1. Prepare stocks in upright T75 flask of P493-6 modified Burkitt's lymphoma cells cultured in tet-free media.
2. On day 0, seed a flask with ~$2 \times 10^7$ P493-6 cells in 40 ml tet-free media with 0.1 µg/ml tet.

   a. Count cells in each flask using a C-chip disposable hemocytometer.

3. 72 hr later, spin cells down from both flasks, and wash cells three times in tet-free media.
4. Seed three new flasks with $1.2 \times 10^7$ cells in 20 ml of fresh tet-free media.

   a. Count cells using a C-chip disposable hemocytometer.

5. At time points 0 hr (immediately), 1 hr, and 24 hr, harvest one flask, and prepare an aliquot of $1 \times 10^7$ cells.

   a. Count cells in each flask using a C-chip disposable hemocytometer.
   b. Homogenize sample in 1 ml Tri Reagent according to manufacturer's instructions.
   c. Store at −80°C until further processing.
   d. Do not exceed $1 \times 10^7$ cells per 1 ml Tri Reagent.

6. For each sample, add 1/10 vol (100 µl per 1 ml Tri Reagent) miRNA Homogenate Additive, vortex, and incubate on ice for 10 min.
7. Add 100 µl bromochloropropane per 1 ml Tri Reagent, vortex for 15–30 s, incubate the homogenate for 5 min at RT, then centrifuge at 12,000×*g* for 10 min at 4°C.

a. Avoid using chloroform containing isoamyl alcohol or water soluble organic solvents (ethanol, DMSO) as this will reduce DNA contamination downstream.

8. Recover the aqueous phase and proceed directly to step F.I (Total RNA Isolation Procedure) in the miRVana isolation kit manufacturer's instructions.

   a. Expect to take 312 µl of aqueous phase, add 390 µl of 100% ethanol, and place the total volume (702 µl) onto the filter cartridge.
   b. Resuspend RNA in 100 µl of nuclease-free water for a total concentration of 100,000 cells/µl.

9. Quantify RNA concentrations in each sample using a spectrometer.

   a. Convert total RNA to ng per 1000 cells.
   b. Record sample purity ($A_{260/280}$ and $A_{260/280}$ ratios).

10. Repeat independently two additional times.

## Deliverables

- Data to be collected:

  1. STR profile and result of mycoplasma testing of P493-6 cells.
  2. Nanodrop measurements of RNA concentrations in total RNA preparations for each sample. Include $A_{260/280}$ and $A_{260/230}$ ratios.
  3. Bar graph of total RNA (ng) per 1000 cells for each condition. (Compare to Figure 3E).

## Confirmatory analysis plan

This experiment assesses the relative levels of total RNA before and after re-induction of c-Myc. The original paper reported an increase in levels of absolute RNA over the timecourse of c-Myc re-induction. This replication attempt will perform the following statistical analysis listed below for each cohort.

- Statistical analysis:

  Note: At the time of analysis we will perform the Shapiro–Wilk test and generate a q–q plot to assess the normality of the data and also perform Levene's test to assess homoscedasiticity. If the data appear skewed we will perform an appropriate transformation in order to proceed with the proposed statistical analysis. If this is not possible we will perform the equivalent non-parametric test.

  1. One-way ANOVA comparing total RNA (ng/1000 cells) in cells cultured 0 hr, 1 hr, and 24 hr from tet release.

     a. Planned comparisons:

        i. Total RNA in cells cultured 0 hr from tet release compared to cells cultured 24 hr from tet release.

- Meta-analysis of effect sizes:

  1. Compare the effect sizes of the original data to the two cohorts (separate FBS lots) and use a meta-analytic approach to combine the original and replication effects, which will be presented as a forest plot.

## Known differences from the original study

A separate lot of serum was included to assess if there is variability between different batches of serum. All known differences are listed in the materials and reagents section above, with the originally used item listed in the comments section. All differences have the same capabilities as the original and are not expected to alter the experimental design.

## Provisions for quality control

The cell line used in this experiment will undergo STR profiling to confirm its identity and will be sent for mycoplasma testing to ensure there is no contamination. The level of c-Myc re-induction in this

system will be established in Protocol 1 as a comparison to the original paper. The sample purity ($A_{260/280}$ and $A_{260/230}$ ratios) of the isolated RNA from each sample will be reported. All of the raw data will be uploaded to the project page on the OSF (https://osf.io/mokeb/) and made publically available.

## Protocol 3: transcript levels during c-Myc re-activation

This experiment, which includes Protocol 3 and 4, tests the effect of elevated levels of c-Myc on the mRNA levels/cell for a large set of genes in P493-6 cells. The data were analyzed to test the effect of c-Myc on transcript levels of genes that were expressed at reliable levels (>1 transcript/cell) or genes that were not expressed (<0.5 transcript/cell). This is a replication of the data presented in Figure 3F and Table S1, which assess the quantified mRNA levels/cells by digital gene expression (NanoString).

## Sampling

- Each experiment has two cohorts:

  1. Cohort 1: P493-6 cells cultured in FBS lot #1.
  2. Cohort 2: P493-6 cells cultured in FBS lot #2.

- Each cohort has three conditons:

  1. Cells cultured 0 hr after tet induction.
  2. Cells cultured 1 hr after tet induction.
  3. Cells cultured 24 hr after tet induction.

- Experiment to be repeated a total of three times.

  1. This replication attempt will be repeated three times to obtain averaged values of transcript/cell since it will be repeated three times in the other replication protocols.

## Materials and reagents

| Reagent | Type | Manufacturer | Catalog # | Comments |
| --- | --- | --- | --- | --- |
| P493-6 modified Burkitt's lymphoma cells | Cell line | Original lab | | From original lab |
| RPMI 1640 medium with sodium bicarbonate, without L-glutamine | Cell culture | Sigma–Aldrich | R0883 | No catalog # listed in original paper |
| Tetracycline system approved FBS | Serum | Clontech | 631105 | Two separate lots will be used |
| Ala-Gln (L-glutamine substitute) | Cell culture | Sigma–Aldrich | G8541 | Original lab used GlutaMAX from Invitrogen |
| T75 flask | Labware | Sigma–Aldrich | Z707503 | Originally not specified |
| Tetracycline | Pharmacological agent | Sigma–Aldrich | T7660 | |
| C-chip disposable hemocytometer | Labware | Digital Bio | DHC-N01 | No catalog # listed in original paper |
| Refrigerated centrifuge | Equipment | Eppendorf | 5810R | Original lab used a Sorvall Legend centrifuge |
| Buffer RLT | Buffer | Qiagen | 79216 | Original lab used the Buffer RLT from the RNeasy kit (#74104) |
| ß-mercaptoethanol | Chemicals | Sigma–Aldrich | M3148 | Included during communication with original authors |
| Nuclease-free water | Chemicals | Sigma–Aldrich | W4502 | Original lab used nuclease-free water from Ambion (AM9938) |
| NanoDrop UV/Vis Spectrophotometer | Equipment | Thermo Scientific | ND-1000 | |
| NanoDrop Operating | Software | Thermo Scientific | Version 3.3 | |

## Procedure

- All cells will be sent for mycoplasma testing and STR profiling.
- Tet-free media: RPMI1640 supplemented with 10% tet system approved FBS and 1% Ala-Gln.
- Cells maintained at 37°C in a humidified atmosphere at 5% $CO_2$.
- P493-6 cells are maintained in two separate FBS lots (lot #1 and lot #2).

1. Prepare stocks of upright T75 flask of P493-6 modified Burkitt's lymphoma cells cultured in tet-free media.
2. On day 0, seed a flask with ~$2 \times 10^7$ P493-6 cells in 40-ml tet-free media with 0.1 µg/ml tet.

   a. Count cells in each flask using a C-chip disposable hemocytometer.
   b. This is performed for cells grown in both lots of FBS.

3. 72 hr later, spin cells down from both flasks, and wash cells three times in tet-free media.
4. Seed three new flasks with $1.2 \times 10^7$ cells in 20 ml of fresh tet-free media.

   a. Count cells using a C-chip disposable hemocytometer.

5. At time points 0 hr (immediately), 1 hr, and 24 hr, harvest one flask, and collect $1 \times 10^6$ cells.

   a. Count cells in each flask using a C-chip disposable hemocytometer.
   b. Lyse cells directly in 100 µl Buffer RLT supplemented with ß-mercaptoethanol (10 µl ß-ME per 1 ml Buffer RLT) for a concentration of 10,000 cells per µl.
   c. Make multiple 4 µl aliquots and store RNA at −80°C until shipment. Avoid freeze/thaw cycles. Ship one aliquot/sample and store others for backup.

6. Repeat independently two additional times.
7. Ship cell lysate samples on dry ice to lab for NanoString nCounter gene expression assay (Protocol 4).

## Deliverables

- Data to be collected:

1. STR profile and result of mycoplasma testing of P493-6 cells.

- Sample delivered for further analysis:

1. Cell lysate samples for digital gene expression assay (Protocol 4).

## Confirmatory analysis plan

This experiment assesses if c-Myc re-induction alters the number of transcripts/cell in genes that were active (>1 transcript/cell) or silent (<0.5 transcript/cell) in cells with lower levels of c-Myc (0 hr after tet induction). The original paper reported an upregulation of active genes upon c-Myc re-induction, while genes not expressed remained silent. This replication attempt will perform the statistical analysis listed in Protocol 4 for each cohort.

## Known differences from the original study

A separate lot of serum was included to assess if there is variability between different batches of serum. All known differences are listed in the materials and reagents section above, with the originally used item listed in the comments section. All differences have the same capabilities as the original and are not expected to alter the experimental design.

## Provisions for quality control

The cell line used in this experiment will undergo STR profiling to confirm its identity and will be sent for mycoplasma testing to ensure there is no contamination. The level of c-Myc re-induction in this system will be established in Protocol 1 as a comparison to the original paper. All of the raw data will be uploaded to the project page on the OSF (https://osf.io/mokeb/) and made publically available.

## Protocol 4: NanoString nCounter digital gene expression assay

This experiment, which includes Protocol 3 and 4, tests the effect of elevated levels of c-Myc on the mRNA levels/cell for a large set of genes in P493-6 cells. The data were analyzed to test the effect

of c-Myc on transcript levels of genes that were expressed at reliable levels (>1 transcript/cell) or genes that were not expressed (<0.5 transcript/cell). This is a replication of the data presented in Figure 3F and Table S1, which assess the quantified mRNA levels/cells by digital gene expression (NanoString).

## Sampling

- Each experiment has two cohorts:

  1. Cohort 1: P493-6 cells cultured in FBS lot #1.
  2. Cohort 2: P493-6 cells cultured in FBS lot #2.

- Each cohort has three conditons:

  1. Cells cultured 0 hr after tet induction.
  2. Cells cultured 1 hr after tet induction.
  3. Cells cultured 24 hr after tet induction.

- Experiment will analyze transcript levels from 1409 unique genes for a minimum power of 80%.

  1. Based on the estimation of detecting 795 active genes and 541 silent genes, which is 56.5% and 38.4%, respectively, of total unique genes, similar to the original data.
  2. See Power Calculations section for details.

## Materials and reagents

| Reagent | Type | Manufacturer | Catalog # | Comments |
|---|---|---|---|---|
| nCounter Custom CodeSet (CS-1, CS-2, CS-MYC combined) | Probe | NanoString | | See Table S1 of original paper for RefSeq IDs |
| nCounter GX Human Immunology Kit | Probe | NanoString | GXA-HIM1-12 | |
| nCounter GX Human Kinase Kit | Probe | NanoString | GXA-P2K1-12 | |
| nCounter Analysis System | Equipment | NanoString | NCT-PREP-120 | Includes Prep Station and Digital Analyzer |
| nSolver Analysis | Software | NanoString | Version 1.1 | |

## Procedure

1. Process samples (from Protocol 3) according to manufacturer's instructions for 'Cell Lysate Protocol' (nCounter Gene Expression Assay Manual).

   a. Thaw samples on ice, mix well, and briefly spin down contents of the tubes.
   b. Incubate 4 µl of cell lysate overnight at 65°C in nCounter Reporter CodeSet, Capture Probe-Set and hybridization buffer.
   c. Following hybridization, process samples immediately with the nCounter PrepStation and subsequently analyze on an nCounter Digital Analyzer.
   d. Approximately 40,000 cells/hybridization assay.

2. Analyze data for all target genes in all samples/flow cells using the nSolver Analysis Software.

   a. Normalize the counts for all target genes in all samples/flow cells based on the positive spike-in controls to account for differences in hybridization efficiency and post-hybridization processing.

      i. Sum the counts for the positive spike-in controls for each sample/flow cell to estimate overall assay efficiency.
      ii. Calculate a normalization factor for each sample/flow cell based on the relative number of positive control counts.
      iii. For each sample/flow cell, multiply the counts for each target gene and the control genes by the normalization factor for that sample/flow cell.

b. Calculate the average normalized background count for a sample/flow cell from the average of the eight negative control counts. Subtract this value from the normalized count for each gene target in that sample/flow cell to yield normalized, background-subtracted counts for each target gene.

c. Average triplicates for each sample.

d. For each gene, estimate its transcripts per cell level through linear interpolation of RNA spike-in positive controls using the approximation that the signal detected from a 0.5 femtomolar RNA spike in is equivalent to the signal detected for a transcript expressed at the equivalent of 1 transcript/cell.

## Deliverables

- Data to be collected:

  1. RCC data output files from nCounter Digital Analyzer before and after normalization, background subtraction, and approximation of transcripts/cell for each gene. Include positive and negative controls for each sample/flow cell.
  2. List of all genes with the transcript/cell estimations for each individual sample and the average across the triplicate samples. (Compare to Table S1).
  3. Box-and-whisker plot of transcript/cell estimates for the active (>1 transcript/cell) and silent (<0.5 transcript/cell) genes at 0, 1, and 24 hr. (Compare to Figure 3F).

## Confirmatory analysis plan

To determine if the normalized, background-subtracted counts are statistically above background, a *t*-test or Wilcoxon rank-sum test, will be used to test against the average counts of the negative control genes. Genes are defined as transcriptionally active (if average expression across replicates is >1 transcript/cell at 0 hr) or transcriptionally silent (if average expression across replicates is <0.5 transcript/cell). Some genes are represented on multiple nCounter Reporter CodeSets, so the average expression will be computed from replicates of all CodeSets with the same RefSeq ID. Genes with an average expression between 0.5 and 1 transcript/cell at 0 hr will be excluded from the analysis of silent genes, similar to how the original paper analyzed the data, but will be included in an additional analysis of non-active genes.

This experiment assesses if c-Myc re-induction alters the number of transcripts/cell in genes that were active (>1 transcript/cell) or silent (<0.5 transcript/cell) in cells with lower levels of c-Myc (0 hr after tet induction). The original paper reported an upregulation of active genes upon c-Myc re-induction, while genes not expressed remained silent. This replication attempt will perform the following statistical analysis listed below.

- Statistical analysis:

  1. Two-tailed Wilcoxon signed rank test of transcript levels of active genes in cells for the following comparisons with the Bonferroni correction:

     a. Cultured 0 hr from tet release compared to 1 hr from tet release.
     b. Cultured 0 hr from tet release compared to 24 hr from tet release.
     c. Cultured 1 hr from tet release compared to 24 hr from tet release.

  2. Two-tailed Wilcoxon signed rank test of transcript levels of silent genes in cells for the following comparisons with the Bonferroni correction:

     a. Cultured 0 hr from tet release compared to 1 hr from tet release.
     b. Cultured 0 hr from tet release compared to 24 hr from tet release.
     c. Cultured 1 hr from tet release compared to 24 hr from tet release.

- Meta-analysis of effect sizes:

  1. Compare the effect sizes of the original data to the two cohorts (separate FBS lots) and use a meta-analytic approach to combine the original and replication effects, which will be presented as a forest plot.

## Known differences from the original study

The original study used three separate custom code sets (CS-1, CS-2, and CS-MYC) and the replication will combine these into one custom code set. The three original custom code sets had a total of 495 genes, of which only 479 were unique, so the replication custom code set will only contain these unique genes. The nCounter GX human immunology kit from NanoString has been updated to a new version. In the second version, there are 106 new genes not present in the first version, of which 101 are unique to all genes examined in this experiment, while 38 genes that were in the first version are excluded in the second version. Thus, 63 unique genes will be added to the total gene set compared with the original study changing the total examined genes from 1346 to 1409. A separate lot of serum was included to assess if there is variability between different batches of serum. The original study analyzed these data using the Wilcoxon sum rank test, however since expressions of the same gene across different conditions are not independent, the Wilcoxon signed rank test, a non-parametric test for comparing two paired samples, will be used instead.

## Provisions for quality control

Each of the nCounter Reporter CodeSets contain internal positive and negative controls that are used to normalize/plot a regression line to quantify relative levels of expression and to estimate the non-specific background, respectively. Genes will be tested to determine if the average normalized, background-subtracted counts are statistically above background. All of the raw data, including the data output files from the nCounter Digital Analyzer, will be uploaded to the project page on the OSF (https://osf.io/mokeb/) and made publically available.

## Power calculations

### Protocol 1

Power calculations are not applicable.

### Protocol 2

Summary of original data (shared by original authors).

| Data set being analyzed | Mean | SD | N |
|---|---|---|---|
| Total RNA 0 hr from tet release | 4.252 | 0.1617 | 3 |
| Total RNA 1 hr from tet release | 4.037 | 0.2946 | 3 |
| Total RNA 24 hr from tet release | 5.471 | 0.3752 | 3 |

### Test family

- ANOVA: Fixed effects, omnibus, one-way, alpha error = 0.05.

  1. Power calculations were performed with G*Power software (version 3.1.7) (*Faul et al., 2007*).
  2. ANOVA F statistic calculated with Graphpad Prism 6.0.
  3. Partial $\eta^2$ calculated from *Lakens (2013)*.

### Power calculations for replication

| Groups | F test statistic | Partial $\eta^2$ | Effect size f | A priori power | Total sample size |
|---|---|---|---|---|---|
| 0 hr, 1 hr, 24 hr | $F_{(2, 6)} = 21.2183$ | 0.8761 | 2.65914 | 99.9% | 9 (3/group) |

### Test family

- Two-tailed *t*-test, difference between two independent means, alpha error = 0.05.

  1. Power calculations were performed with G*Power software (version 3.1.7) (*Faul et al., 2007*).

## Power calculations for replication

| Group 1 | Group 2 | Effect size $d$ | A priori power | Group 1 sample size | Group 2 sample size |
|---------|---------|-----------------|----------------|---------------------|---------------------|
| 0 hr | 24 hr | 4.19268947 | 96.3% | 3 | 3 |

### Protocol 3 and 4
Summary of original data (obtained from Table S1)

| Data set being analyzed | N | Mean | SD |
|-------------------------|---|------|-----|
| Active genes 0 hr from tet release | 760 | 23.667 | 88.104 |
| Active genes 1 hr from tet release | 760 | 30.418 | 117.36 |
| Active genes 24 hr from tet release | 760 | 54.959 | 179.67 |
| Silent genes 0 hr from tet release | 517 | 0.0646 | 0.1225 |
| Silent genes 1 hr from tet release | 517 | 0.0837 | 0.2810 |
| Silent genes 24 hr from tet release | 517 | 0.2981 | 1.3787 |

Note: Active genes and silent genes are defined as >1 transcript/cell and <0.5 transcript/cell as in the original paper.

### Test family

- Two-tailed Wilcoxon signed rank test, alpha error = 0.01667.
  1. Power calculations were performed with G*Power software (version 3.1.7) (*Faul et al., 2007*).

### Power calculations for replication (active genes)

| Group 1 | Group 2 | Effect size $d_z$ | A priori power | Total sample size |
|---------|---------|-------------------|----------------|-------------------|
| 0 hr from tet release | 1 hr from tet release | 0.2231897 | 80.2%* | 224* |
| 0 hr from tet release | 24 hr from tet release | 0.3031793 | 80.3%* | 123* |
| 1 hr from tet release | 24 hr from tet release | 0.2898147 | 80.2%* | 134* |

*795 is used based on the estimation of 56.5% of total unique genes being active making the power 99.9%.

### Test family

- Two-tailed Wilcoxon signed rank test, Bonferroni's correction, alpha error = 0.01667.
  1. Power calculations were performed with G*Power software (version 3.1.7) (*Faul et al., 2007*).

### Power calculations for replication (silent genes)

| Group 1 | Group 2 | Effect size $d_z$ | A priori power | Total sample size |
|---------|---------|-------------------|----------------|-------------------|
| 0 hr from tet release | 1 hr from tet release | 0.1427513* | 80.0%* | 541* |
| 0 hr from tet release | 24 hr from tet release | 0.1743707 | 80.1%† | 364† |
| 1 hr from tet release | 24 hr from tet release | 0.1686160 | 80.0%‡ | 389‡ |

*This is a sensitivity calculation. The original effect size is 0.0807517.
†541 is used based on the estimation of 38.4% of total unique genes being silent making the power 94.0%.
‡541 is used making the power 92.3%.

## Acknowledgements

The Reproducibility Project: Cancer Biology core team would like to thank the original authors, in particular Charles Lin, for generously sharing critical information as well as the P493-6 modified Burkitt's lymphoma cells to ensure the fidelity and quality of this replication attempt. We thank Courtney Soderberg at the Center for Open Science for assistance with statistical analyses. We would also like to thank the following companies for generously donating reagents to the Reproducbility Project: Cancer Biology; American Type Culture Collection (ATCC), BioLegend, Cell Signaling Technology, Charles River Laboratories, Corning Incorporated, DDC Medical, EMD Millipore, Harlan Laboratories, LI-COR Biosciences, Mirus Bio, Novus Biologicals, Sigma–Aldrich, and System Biosciences (SBI).

## Additional information

### Group author details

**Reproducibility Project: Cancer Biology**

Elizabeth Iorns: Science Exchange, Palo Alto, California; William Gunn: Mendeley, London, United Kingdom; Fraser Tan: Science Exchange, Palo Alto, California; Joelle Lomax: Science Exchange, Palo Alto, California; Timothy Errington: Center for Open Science, Charlottesville, Virginia

### Competing interests

DB: This is a Science Exchange Associated lab. HH: This is a Science Exchange Associated lab. RP:CB: EI, FT and JL are employed by and hold shares in Science Exchange Inc. The other author declares that no competing interests exist.

### Funding

| Funder | Author |
| --- | --- |
| Laura and John Arnold Foundation | Reproducibility Project: Cancer Biology |

The Reproducibility Project: Cancer Biology is funded by the Laura and John Arnold Foundation, provided to the Center for Open Science in collaboration with Science Exchange. The funder had no role in study design or the decision to submit the work for publication.

### Author contributions

DB, HH, MMC, Drafting or revising the article; RP:CB, Conception and design, Drafting or revising the article

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
