## [Decision Letter]

Thank you for sending your work entitled “Registered report: Transcriptional
Amplification in Tumor Cells with Elevated *c-Myc*” for
consideration at *eLife*. Your article has been evaluated by Sean
Morrison (Senior editor), a Reviewing editor, and 3 reviewers, one of whom is a
biostatistician.

The Reviewing editor and the reviewers discussed their comments before we reached this
decision, and the Reviewing editor has assembled the following comments to help you
prepare a revised submission.

As detailed below, the reviewers raised a number of major concerns that need to be
addressed in a revised Registered report.

Major comments:

1) Any replication of the Lin et al. paper needs to include both RNA profiling and
ChIP-seq experiments. The Lin et al. paper has been quite controversial and was based in
very large part on ChIP-seq data and its interpretation. Many of the conclusions were
based on very subtle changes in data profiles. Subsequently, two papers were published
in Nature (Walz et al. and Sabo et al.) that challenge the global claims of Lin et al.
In both papers, the authors worked hard to accumulate comprehensive ChIP-seq and RNA
expression data in carefully designed experimental systems. While some aspects of the
Lin et al. paper may be correct, both Nature papers conclude that there is a set of
defined target genes that are far more *Myc* responsive than others.
Hence, reproducing only a subset of the Lin et al. experiments is unlikely to add
anything new or resolve controversial claims.

The authors do not propose to reproduce the critical ChIP-seq data and they do not
propose any analysis of RNAPII profiles that would support or conflict with the
conclusions of the Lin et al. paper, namely that *Myc* promotes
genome-wide transcriptional elongation. The proposal only focuses on RNA profiling
without integration with binding of *Myc* and RNAPII.

2) More comprehensive RNA-seq analysis would determine global RNA expression in response
to *Myc* and not be limited to a subset of genes represented by
NanoString.

3) The cell line P493-6 has been established 15 years ago. The proliferation of these
cells depends on *c-Myc* expression *and* the presence of
serum. Serum is a major variable in this system and the majority of genes in stimulated
cells are regulated by *serum* and not by *c-Myc*
(Schlosser et al., Oncogene, 2005). The impact of different serum changes on P493-6
cells is highly significant. In some serum batches the cells barely grow after
*c-Myc* activation. Unfortunately, the impact of various serum batches
on P493-6 cells has never been systematically analyzed. Moreover, meanwhile many batches
of P493-6 cells are distributed worldwide. These cells have been cultured with different
types of serum in different laboratories. Exposure to different sera probably has
altered the epigenetic state of P493-6 cells further contributing to variation in gene
expression.

From the scientific view, it would be more helpful to study the stability of this
biological system, e.g. by culturing P493-6 cells over longer periods of time in
different batches of serum followed by a subsequent transcriptome analysis +/-
*Myc*. At minimum, the authors should perform their experiments using
multiple batches of serum to assess whether this significantly alters their results.

Statistical comments to the authors:

4) For protocol 1 and 2, authors propose to use ANOVA to analyze the data. Please make
sure that the data do not violate the assumptions of the ANOVA: normality and
homoscedasiticity. If the data do not fit the assumptions well enough, try to find a
data transformation that makes them fit. If this doesn't work, you will need to
apply a nonparametric counterpart of ANOVA such as Kruskal-Wallis test. In addition,
performing contrast within the framework of ANOVA is more powerful than performing a
separate t-test if the assumption of ANOVA is valid.

5) Authors used G*Power to calculate the power. I think that power calculation for
protocol 3 & 4 is probably based on the test family t-test implemented in
G*Power since there is no Wilcoxon sum rank test implemented in the G*Power. I
suggest using t-tests as test family and matched pairs as statistical test to
recalculate the power for protocol 3 and 4 (see below for justification). You will need
to re-compute the effect size by calculating the SD for paired design, although mean
difference between two groups will stay the same regardless.

6) Authors propose to use two-tailed Wilcoxon sum rank test, which has been used in the
original paper. I suggest use either two-tailed Wilcoxon signed rank test or two-tailed
paired t-test. If you prefer use G* Power to calculate power, then you will be left
with two-tailed paired t-test option. The reason why paired analysis is needed is that
expressions of the same gene across different conditions are not independent.

7) One major conclusion from the original paper (5) is that elevated *c-Myc* in tumor cells leads to
amplification of the expression of actively transcribed genes, but has no effect on
silent genes. I am wondering whether the authors will perform the same test to the
silent genes, as well as the actively transcribed genes, to confirm the results from the
original paper.

8) While it is very useful to leverage the previously reported effects to compute
minimum power a priori, what you really need is to guarantee a minimum power on your own
data. This can be done, a priori, by including some cross-study variation. This will be
helpful for you to plan on the number of replicates and so forth. Papers by Giovanni
Parmigiani and collaborators at the Dana–Farber provide some estimates about
cross-study variation that could be used for this purpose. Worst case, you should budget
some additional variability because of cross-study reproducibility, and increase the
sample size as appropriate. We also want you to compute and report power
post-hoc/on-the-fly on your own data. Some minimum power should be guaranteed using
summaries of your own data.

[Editors' note: further revisions were requested prior to acceptance, as described
below.]

Thank you for resubmitting your work entitled “Registered report: Transcriptional
Amplification in Tumor Cells with Elevated *c-Myc*” for further
consideration at *eLife*. Your revised article has been favorably
evaluated by Sean Morrison (Senior editor), a Reviewing editor, and the original
reviewers. As you might expect, there was a mixed response from the reviewers regarding
the changes. On balance, we would like to move forward but would ask you to make one
additional change. Different serum batches have only been included for the
*c-Myc* off situation. To complete this control, please also include
different serum batches for the c-*Myc* on situation.

---

## [Author Response]

*1) Any replication of the Lin et al. paper needs to include both RNA profiling
and ChIP-seq experiments. The Lin et al. paper has been quite controversial and was
based in very large part on ChIP-seq data and its interpretation. Many of the
conclusions were based on very subtle changes in data profiles. Subsequently, two
papers were published in Nature (Walz et al. and Sabo et al.) that challenge the
global claims of Lin et al. In both papers, the authors worked hard to accumulate
comprehensive ChIP-seq and RNA expression data in carefully designed experimental
systems. While some aspects of the Lin et al. paper may be correct, both Nature
papers conclude that there is a set of defined target genes that are far
more* Myc *responsive than others. Hence, reproducing only a subset of
the Lin et al. experiments is unlikely to add anything new or resolve controversial
claims.*

*The authors do not propose to reproduce the critical ChIP-seq data and they do
not propose any analysis of RNAPII profiles that would support or conflict with the
conclusions of the Lin et al. paper, namely that* Myc *promotes
genome-wide transcriptional elongation. The proposal only focuses on RNA profiling
without integration with binding of* Myc *and RNAPII.*

We agree that all of the experiments included in the original study are important, and
choosing which experiments to replicate has been one of the great challenges of this
project. The Reproducibility Project: Cancer Biology (RP:CB) aims to replicate
experiments that are impactful, but does not necessarily aim to replicate all the
impactful experiments in any given paper. In this case, the RP:CB core team felt that
the RNA profiling experiment in [5], was critical as it was a key part of the reported finding that
*Myc* induction increased the expression of active genes but not
silent genes, indicating the predominant effect of substantially elevated levels of
*Myc* is amplified transcription of the existing gene expression
program. We also agree the ChIP-seq experiments are a critical component of the overall
reported finding, however these types of experiments (sequencing) are excluded from all
articles. As this was only a part of the paper, the other experiments were still
eligible. These exclusion criteria are outlined on the project page (https://osf.io/e81xl/wiki/studies) and in a Feature article describing
the project (recently accepted for publication by *eLife*). We agree that
the exclusion of certain experiments limits the scope of what can be analyzed by the
project, but we are attempting to identify a balance of breadth of sampling for general
inference with sensible investment of resources on replication projects to determine to
what extent the included experiments are reproducible. Therefore, we will restrict our
analysis to the experiments being replicated and will not include discussion of
experiments not being replicated in this study.

We have also updated the Introduction to include the two papers (Walz et al. and Sabo et
al.) that also examine transcriptional regulation by *Myc*.

*2) More comprehensive RNA-seq analysis would determine global RNA expression in
response to* Myc *and not be limited to a subset of genes represented
by NanoString.*

We agree RNA-seq analysis would be an informative approach to determine the global RNA
expression in response to *Myc*, however Lin and colleagues did not use
this approach. The Reproducibility Project: Cancer Biology aims to perform direct
replications using the same methodology reported in the original paper. The use of
RNA-seq analysis would be a conceptual replication, which we agree is a useful approach
to test the experiment’s underlying hypothesis, but which is not an aim of the
project.

*3) The cell line P493-6 has been established 15 years ago. The proliferation of
these cells depends on* c-Myc *expression* and *the
presence of serum. Serum is a major variable in this system and the majority of genes
in stimulated cells are regulated by* serum *and not by* c-Myc
*(Schlosser et al., Oncogene, 2005). The impact of different serum changes on
P493-6 cells is highly significant. In some serum batches the cells barely grow
after* c-Myc *activation. Unfortunately, the impact of various serum
batches on P493-6 cells has never been systematically analyzed. Moreover, meanwhile
many batches of P493-6 cells are distributed worldwide. These cells have been
cultured with different types of serum in different laboratories. Exposure to
different sera probably has altered the epigenetic state of P493-6 cells further
contributing to variation in gene expression.*

*From the scientific view, it would be more helpful to study the stability of
this biological system, e.g. by culturing P493-6 cells over longer periods of time in
different batches of serum followed by a subsequent transcriptome analysis
+/-* Myc. *At minimum, the authors should perform their
experiments using multiple batches of serum to assess whether this significantly
alters their results.*

This project focuses on direct replication of the experiments as detailed in the
original report and with information provided by the original authors. Aspects of an
experiment not included in the original study are occasionally added to ensure the
quality of the research, but by no means is a requirement of this project; rather, it is
an extension of the original work. Adding additional aspects not included in the
original study can be of scientific interest, and can be included, if it is possible, to
balance them with the main aim of this project: to perform a direct replication of the
original experiment(s).

Therefore, we agree with the reviewers that there is scientific interest in better
understanding the stability of this biological system. We will be using the same P493-6
cells used by the original authors (generously provided by Charles Lin) and will use the
same source of Tet System Approved FBS as originally reported. However, even if we were
to know the lot number of FBS originally used, it is unlikely we would be able to obtain
it. We will use the same lot for all the experiments described, but have included an
additional cohort to all experiments that will use a different lot of FBS. This cohort
will be harvested in the same manner described for the cohort harvested 0 hr after
tetracycline induction. These two cohorts, both 0 hr after tetracycline induction, but
with different lots of FBS, will be compared to each other to assess if there is
variability between different batches of serum.

*Statistical comments to the authors*:

*4) For protocol 1 and 2, authors propose to use ANOVA to analyze the data.
Please make sure that the data do not violate the assumptions of the ANOVA: normality
and homoscedasiticity. If the data do not fit the assumptions well enough, try to
find a data transformation that makes them fit. If this doesn't work, you will
need to apply a nonparametric counterpart of ANOVA such as Kruskal-Wallis test. In
addition, performing contrast within the framework of ANOVA is more powerful than
performing a separate t-test if the assumption of ANOVA is valid*.

Thank you for this suggestion. At the time of analysis, we will perform the Shapiro-Wilk
test and generate a quantile-quantile (q-q) plot, to assess the normality of the data,
and also perform the Brown-Forsythe test to assess homoscedasiticity. If the data
appears skewed, we will perform the appropriate transformation in order to proceed with
the proposed statistical analysis. We will note any changes or transformations made. If
this doesn’t work we will perform the Kruskal-Wallis test and if necessary the
Wilcoxon-Mann-Whitney test. We have also updated the manuscript to address this
point.

Also, in protocol 1 we made the intended planned comparison (contrast) explicit to
clarify that it is not a separate t-test.

*5) Authors used G*Power to calculate the power. I think that power
calculation for protocol 3 & 4 is probably based on the test family t-test
implemented in G*Power since there is no Wilcoxon sum rank test implemented in
the G*Power. I suggest using t-tests as test family and matched pairs as
statistical test to recalculate the power for protocol 3 and 4 (see below for
justification). You will need to re-compute the effect size by calculating the SD for
paired design, although mean difference between two groups will stay the same
regardless*.

G*Power does have the Wilcoxon sum rank test as an option. It is called
Wilcoxon-Mann-Whitney test, which is another name for this test. However, we agree with
the recalculation for protocols 3 and 4 suggested below and now include the two-tailed
Wilcoxon signed rank test instead of the Wilcoxon sum rank test.

*6) Authors propose to use two-tailed Wilcoxon sum rank test, which has been used
in the original paper. I suggest use either two-tailed Wilcoxon signed rank test or
two-tailed paired t-test. If you prefer use G* power to calculate power, then
you will be left with two-tailed paired t-test option. The reason why paired analysis
is needed is that expressions of the same gene across different conditions are not
independent*.

We agree with this assessment. A two-tailed Wilcoxon signed rank test is the appropriate
test because the data is paired and the normality assumption does not hold. We have
recalculated power and sample size for protocols 3 and 4 accordingly. We also changed
the language in the analysis plan and added this as a known difference from the original
study.

*7) One major conclusion from the original paper (*[5]*) is that
elevated* c-Myc *in tumor cells leads to amplification of the
expression of actively transcribed genes, but has no effect on silent genes. I am
wondering whether the authors will perform the same test to the silent genes, as well
as the actively transcribed genes, to confirm the results from the original
paper.*

We agree and have included the analysis of silent genes in the analysis plan and
performed sensitivity calculations to determine the effect size that will be detected
with 80% power. In the original paper (5), the authors determined actively transcribed genes as > 1
transcript/cell and silent genes as < 0.5 transcript/cell in cells with lower
levels of *c-Myc* (0 hr after tet induction). However, this excludes 69
genes from the analysis that fall between 1 and 0.5. Thus, in the analysis plan we have
included silent genes as defined by the original authors (< 0.5 transcript/cell)
and also non-active genes (< 1 transcript/cell).

*8) While it is very useful to leverage the previously reported effects to
compute minimum power a priori, what you really need is to guarantee a minimum power
on your own data. This can be done, a priori, by including some cross-study
variation. This will be helpful for you to plan on the number of replicates and so
forth. Papers by Giovanni Parmigiani and collaborators at the Dana–Farber
provide some estimates about cross-study variation that could be used for this
purpose. Worst case, you should budget some additional variability because of
cross-study reproducibility, and increase the sample size as appropriate. We also
want you to compute and report power post-hoc/on-the-fly on your own data. Some
minimum power should be guaranteed using summaries of your own data*.

We thank the reviewers for these suggestions. The cross-study variation, such as
approaches that utilize the 95% confidence interval of the effect size, can be useful in
conducting power calculations when planning adequate sample sizes for detecting the true
population effect size, which requires a range of possible observed effect sizes.
However, the Reproducibility Project: Cancer Biology is designed to conduct replications
that have 80% power to detect the point estimate of the originally reported effect size.
While this has the limitation of being underpowered to detect smaller effects than what
is originally reported, this standardizes the approach across all studies to be designed
to detect the originally reported effect size with at least 80% power. Also, while the
minimum power guarantee is beneficial for observing a range of possible effect sizes,
the experiments in this replication, and all experiments in the project, are designed to
detect the originally reported effect size with a minimum power of 80%. Thus, performing
power calculations during or after data collection is not necessary in this replication
attempt as all studies included are already designed to meet a minimum power or are
identified beforehand as being underpowered and thus are not included in the
confirmatory analysis plan. The papers by Giovanni Parmigiani and collaborators
highlight the importance of accounting for variability that can occur across different
studies, specifically gene expression data. While it is possible for a difference in
variance between the originally reported results and the replication data, this will be
reflected in the presentation of the data and a possible reason for obtaining a
different effect size estimate.

[Editors’ note: further revisions were requested prior to acceptance, as
described below.]

*Thank you for resubmitting your work entitled ”Registered report:
Transcriptional Amplification in Tumor Cells with Elevated*
c-Myc*” for further consideration at eLife. Your revised article has
been favorably evaluated by Sean Morrison (Senior editor), a Reviewing editor, and
the original reviewers. As you might expect, there was a mixed response from the
reviewers regarding the changes. On balance, we would like to move forward but would
ask you to make one additional change. Different serum batches have only been
included for the* c-Myc *off situation. To complete this control,
please also include different serum batches for the* c-Myc *on
situation.*

We agree and have adjusted the manuscript. We will use two different lots of serum to
grow the cells for all the experiments.